# A Comparison between Male and Female Athletes in Relative Strength and Power Performances

**DOI:** 10.3390/jfmk6010017

**Published:** 2021-02-09

**Authors:** Sandro Bartolomei, Giuseppe Grillone, Rocco Di Michele, Matteo Cortesi

**Affiliations:** 1Department of Biomedical and Neuromotor Sciences, University of Bologna, 40136 Bologna, Italy; giuseppe.grillone@studio.unibo.it (G.G.); rocco.dimichele@unibo.it (R.D.M.); 2Department for Life Quality Studies, University of Bologna, 40136 Bologna, Italy; m.cortesi@unibo.it

**Keywords:** lean body mass, muscle thickness, mid-shin pull, bench press throw

## Abstract

The aim of this study was to compare male vs. female athletes in strength and power performance relative to body mass (BM) and lean body mass (LBM) and to investigate the relationships between muscle architecture and strength in both genders. Sixteen men (age = 26.4 ± 5.0 years; body mass = 88.9 ± 16.6 kg; height = 177.6 ± 9.3 cm) and fourteen women (age = 25.1 ± 3.2 years; body mass = 58.1 ± 9.1 kg; height = 161.7 ± 4.8 cm) were tested for body composition and muscle thickness (MT) of vastus lateralis muscle (VT), pectoralis major (PEC), and trapezius (TRAP). In addition, participants were tested for lower body power at countermovement jump (CMJP) and upper-body power at bench press throw (BPT). Participants were also assessed for one repetition maximum (1RM) at bench press (1RMBP), deadlift (1RMDE), and squat (1RMSQ). Significantly greater (*p* < 0.01) MT of the VL, PEC and TRAP muscles and LBM were detected in men compared to women. Significantly greater (*p* < 0.05) 1RMBP and BPT adjusted for LBM were detected in men than in women. No significant gender differences after adjusting for LBM were detected for 1RMSQ (*p* = 0.945); 1RMDE (*p* = 0.472) and CMJP (*p* = 0.656). Significantly greater (*p* < 0.05) results in all performance assessments adjusted for MT of the specific muscles, were detected in males compared to females. Superior performances adjusted for MT and LBM in men compared to women, may be related to gender differences in muscle morphology and LBM distribution, respectively.

## 1. Introduction

Women athletes are known to be less strong and powerful than equally trained men [1], muscle strength of women indeed, is typically reported in the range of 40 to 75% of that of men [2]; women are also known to be less powerful than equally trained men. [3]. Gender differences are still evident when power per kg of body mass is considered [3,4] and the difference in absolute strength between genders appears more evident in the upper body compared to the lower body [5,6]. On the contrary, several studies have reported that strength per unit of cross sectional area or lean body mass, do not substantially differ between sexes [7,8,9]. The main factors accounting for gender differences in maximal strength, indeed, have been identified as the muscle mass [6,7]. Other studies confirmed that gender differences in strength may be accounted to LBM but reported that the differences in power performances were still apparent regardless of body composition, and muscle mass [10,11]. These results support the idea that differences between genders in anaerobic power and jumping capacity could not be accounted for by differences in lean body mass only [10,11].

Despite no significant differences between genders in muscle fibers number were reported [6], a qualitative difference in muscle tissue, such as a higher concentration of glycolytic enzymes and greater proportion of fast type muscle fibers [12], may explain the disparity in strength. Glycolytic capacity [12,13], as well as the muscle area occupied by fast type fibers, indeed, have been reported to be greater in male than in female individuals [14]. The gender difference in power may be also influenced by anthropometric and task-specific factors as well as morphological characteristics of muscles. More recently however, Perez-Gomez et al. [15], found similar power output normalized to muscle mass of the lower limbs in males and females. Maximal strength and power are influenced by many neuromuscular factors including muscle morphological characteristics such as muscle thickness, pennation angle, and fascicle length [16,17]. Recently, moderate correlations have been reported between maximal isometric force expressed at the mid-shin pull and muscle architecture of the vastus lateralis in resistance-trained individuals [16]. In addition, significant correlations were detected between muscle architecture of the vastus intermedius and the late phase of the rate of force development at isometric leg extension [18]. Further studies also reported significant correlations between muscle architecture of the vastus lateralis and peak power and time to peak power in an all-out Wingate test [19]. These parameters of muscle architecture, may be influenced by resistance training in both male and female athletes [20,21].

To the best of our knowledge, no studies to date have compared strength and power performances of male and female athletes in relation to body composition and muscle architecture. Therefore, the aim of this study was to compare male vs. female athletes in strength and power performance relative to body mass, lean body mass, and muscle architecture. A second aim of the present study was to assess the relationships between muscle architecture and maximal strength in resistance trained athletes of both genders competing in sports with high expressions of strength and power. The research hypothesis was that higher levels of maximum strength and power relative to body mass, lean body mass and muscle thickness may be found in male compared to women [10,11].

## 2. Materials and Methods

### 2.1. Participants

The participants in the present study were 16 men (M: age = 26.4 ± 5.0 years; body mass = 88.9 ± 16.6 kg; height = 177.6 ± 9.3 cm) and 14 women (W: age = 25.1 ± 3.2 years; body mass = 58.1 ± 9.1 kg; height = 161.7 ± 4.8 cm) athletes competing in strength and power events such as weightlifting (3 males and 3 females), rugby union (9 males and 7 females), and track and field (throwing disciplines and pole vault: 4 males and 4 females). The participants were strength trained at least three times per week (mean = 4.6 ± 1.9 and 4.1 ± 1.5 workouts/week in men and women, respectively), for more than three years (mean = 5.2 ± 2.9 and 4.1 ± 2.4 years of experience in men and women, respectively), and were familiar with both powerlifting and weightlifting exercises. Inclusion criteria required the participants to have competed at regional level in the year previous to the study. Only two athletes of male group and 2 athletes of female group have competed in weightlifting national events in the year previous to the study. Exclusion criteria included injuries which occurred in the year before the study. All the participants volunteered to take part in the present study and signed an informed consent document after having the risks and benefits of the study explained to them. The participants were asked to abstain from alcohol, caffeine, any dietary supplement, and resistance training for at least 24 h prior to the tests. The study was approved by the local University bioethics committee.

### 2.2. Design and Methodologies

In the present cross-sectional, descriptive, and correlational study, participants reported to the laboratory on two separate occasions (see Figure 1), 48 h apart. In the first visit they were assessed for anthropometric measurement, for upper and lower body power and for maximal isometric strength. In addition, the participants performed a squat 1 repetition maximum (1RMSQ). In the second visit the participants were assessed for muscle architecture of the vastus lateralis (VL), superior part of trapezius muscle (TRAP), pectoralis major (PEC), and were asked to perform a bench press and a deadlift 1RM test (1RMBP and 1RMDE, respectively). The estimated sample size was 20 to detect a between-group difference of 200 w and of 30 kg in upper body power and strength, respectively.

### 2.3. Strength and Power Testing

Anthropometric evaluations were performed at the beginning of the first assessment session, and included body mass, height, and body fat composition. Body mass was measured to the nearest 0.1 kg using a scale (Seca 769, Seca Scale Corp., Munich, Germany). Skinfold caliper measures were performed following the method proposed by Evans et al. [22]. All measurements were performed by the same investigators using an Harpender Skinfold Caliper (Harpenden, British Indicators, West Sussex, UK). Prior to the strength and power assessments, the participants performed a standardized warm-up consisting of five min on a cycle ergometer against a light resistance, 10 body weight squats, 10 body weight walking lunges, 10 dynamic walking hamstring stretches, 10 dynamic walking quadriceps stretches, and 5 push-ups [23]. The participants were assessed for upper body power using a bench press throw (BPT) test performed on a Smith Machine as previously described by Bartolomei et al. [24]. The participants pressed loads corresponding to 50% of their estimated 1RM (based on the loads used in training). Two sets of 2 repetitions were required with a recovery time of 20 s between repetitions and of 3 min between sets. During all repetitions, an optical encoder (Tendo Unit model V104; Tendo Sports Machines, Trencin, Slovak Republic) was used to measure the mean velocity and to calculate the power generated using the following equation: Mean Power (W) = repetition mean speed (m·s^−1^) x force (N). Then, the participants were tested for countermovement jump test (CMJ) using a contact mat (Globus Ergo Jump, Globus Ent. Codognè, Italy). They were required to keep their hands on their hips and to maximize their jump height, as previously described by Coratella et al. [25]. They were also asked to perform 3 jumps with a recovery time of 3 min between the attempts. Peak power (CMJP) was calculated by the jump height and the participant’s body mass using the following equation: Peak Power = 60.7 × jump height + 45.3 × body mass − 2055 [26]. In both CMJP and BPT assessments, the best trials were registered and used for analysis.

Isometric maximal strength assessments consisted of an isometric mid-shin pull (MSP) test performed in randomized order on a power rack that permitted fixation of the bar at a height that corresponded to the participant’s mid-shin while standing on a force plate (Kistler Force Plate, Winterthur, Switzerland, 500 Hz). For MSP, bar was set at a distance of 22.5 cm from the floor to the center of the bar to reproduce the official bar height in weightlifting and powerlifting competitions [16]. During MSP, the participants were secured to the bar using lifting straps and subsequently performed 2 maximal isometric pulls lasting for 6 s with a recovery time of 3 min between attempts. For MSP, peak force (PF) was measured. Intraclass coefficients were 0.94 (SEM = 158.4 N) and 0.99 (SEM = 32.56) for PF at MSP.

Bench press 1RM, squat 1RM and deadlift 1RM tests were performed as described in previous studies [16]. Briefly, each participant was asked to perform two warm-up sets using 40–60 and 60–80% of their perceived 1RM, respectively. Then, 3–4 subsequent trials were performed to determine 1RM. The rest period between each trail was set at 3–5 min. The deadlift 1RM assessment was performed using an Olympic bar and plates (Pallini Sport Inc., Malaquis, France). The same barbell and plates were used for squat and bench press 1RMs. During the squat 1RM, participants were asked to unrack the barbell and squat from a standing position, until the greater trochanter of the femur was at the same level of the knee. An investigator monitored the participant’s technique while another researcher monitored the depth of the squat. In the bench press 1RM, a flat-back technique with feet on the ground was used, and participants were required to lower the bar to their chest before initiating the concentric movement. Then, the participants pushed the bar until his arms were fully extended. Their grip widths were recorded to reproduce the same hands position in all the attempts. A 3-min recovery time was set between 1RM attempts. The participants were required to reach their maximum load within 5 attempts. During all strength measurements, they were verbally encouraged by the study investigators to exert maximum effort in each attempt. In addition, a volitional movement tempo was adopted by the participants in each 1RM attempt [27]. Familiarization trials were not performed because all the participants were already familiar with the performed assessments.

### 2.4. Ultrasonography Measurements

Non-invasive skeletal muscle ultrasound images were collected from the participant’s right side of the body. Prior to image collection, all anatomical locations of interest were identified using standardized landmarks for the vastus lateralis (VL), pectoralis major (PEC), and trapezius (TRAP). These muscles were selected because previous studies indicated a good reliability of the measurements in resistance trained individuals [28]. VL muscle thickness (VLMT) and pennation angle (VLPA) measurements required the participant to lay on their side on the examination table with a 10° bend angle in the knees for a minimum of 15 min before images were collected. The landmark for the VL was identified along its longitudinal distance at 50% from the proximal insertion of the muscle. The length of the VL encompassed the distance from the lateral condyle of the tibia to the most prominent point of the greater trochanter of the femur [16]. For the measurement of muscle thickness of the pectoralis (PECMT), the participant lay in supine position and the site between the third and fourth costa under the clavicle midpoint [29] is identified. Muscle thickness of the trapezius (TRAPMT) was assessed with the participant lay in a prone position with the head in the midline and the arms positioned by their sides with the palms facing the ceiling. TRMT was measured at the midpoint of the muscle belly between T1 and the posterior acromial edge, where the muscle borders were parallel [30]. The same investigator, blinded to treatment allocation, performed all landmark measurements for each participant. A 12 MHz linear probe scanning head (Echo Wave 2, Telemed Ultrasound Medical System, Milan, Italy) was coated with water soluble transmission gel to optimize spatial resolution and used to collect all ultrasound images. The probe was positioned on the surface of the skin without depressing the dermal layer and the view mode (gain = 50 dB; image depth = 5 cm) was used to take pictures of the muscle. All images were taken and analyzed by the same technician. Muscle thickness and VLPA were quantified in still images using the measuring features of the ultrasound device. MT was determined as the distance between subcutaneous adipose tissue-muscle interface and intermuscular interface, and VLPA was determined as the angles between the echoes of the deep aponeurosis of the muscle and the echoes from interspaces among the fascicles. Muscle thickness was determined as the distance between subcutaneous adipose tissue-muscle interface and intermuscular interface, and PA was determined as the angles between the echoes of the deep aponeurosis of the muscle and the echoes from interspaces among the fascicles. Fascicle length was calculated from MT and PA using the following equation [31]:FL = MT × SIN (PA)^−1^

### 2.5. Statistical Analysis

A Shapiro–Wilk test was used to test the normal distribution of the data. A relative reliability index (intra-class correlation coefficient, ICC) was used to examine the level of agreement between the attempts performed in each assessment. Absolute reliability (standard error of measurement, SEM) was used to define the extent to which a score varies on test–retest measurements. Independent sample *t* tests were used to compare the mean values of body composition and muscle architecture variables between groups. In addition, Edge’s g effect size (ES), and 95% confidence intervals (CI) were reported. Analysis of covariance (ANCOVA) was used to compare results of examined parameters between sexes while holding BM, LBM, or MT, constant [12]. For effect size, the partial eta squared (η^2^) was reported, and according to Stevens [32], 0.01, 0.06, and 0.14 represent small, medium, and large effect sizes, respectively. Pearson’s product moment correlations were used to examine selected bivariate relationships. According to Mukkaka et al. [33], correlation coefficients (*r*) of 0.3, 0.5, 0.7, and 0.9 were interpreted as low, moderate, high, and very high, respectively. Significance was accepted at an alpha level of *p* ≤ 0.05, and all data are reported as mean ± SD. All analyses were performed using IBM SPSS, version 25.

## 3. Results

All the data relative to performance assessments and muscle architecture were normally distributed (*p* < 0.05). Intraclass coefficients of correlation were 0.94 (SEM = 158.4 N; CV = 0.22), 0.99 (SEM = 32.56; CV = 0.19) and 0.96 (SEM = 100.3 w; CV = 0.20) for PF at MSP, BPT and CMJP, respectively. Intra-class correlation coefficients (ICCs) were 0.96 (SEM = 0.63 mm); 0.93 (SEM = 1.1°) and 0.96 (SEM = 8.0 mm) for muscle thickness MT, pennation angle (PA) and fascicle length (FL) of vastus lateralis, respectively. Intra-class correlation coefficients were 0.97 (SEM: 0.52 mm) and 0.98 (SEM: 0.41 mm) for the muscle thickness of PEC and TRAP, respectively.

Anthropometric data of both groups are reported in Table 1.

Significant differences between men and women were detected for lean body mass (LBM) (*p* < 0.001; ES = 3.234; CI: 23.99, 38.54) while no significant differences were noted for fat mass (FM) (*p* = 0.721; ES = 0.606; CI: −3.78, 5.40). LBM was 41.3% lower in men than in women. Ultrasound measurements were significantly different between groups for muscle thickness of TRAP (*p* < 0.001; ES = 2.158; CI: 0.33, 0.69; −36.7% in women compared to men), PEC (*p* < 0.001; ES = 3.203; CI: 0.79, 1.29; −45.4% in women compared to men), VL (*p* < 0.001; ES = 1.637; CI: 0.30, 0.82; −27.3% in women compared to men), and for the fascicle length of VL (*p* = 0.002; ES = 1.506; CI: 1.31, 5.30; −27.3% in women compared to men). No significant differences were noted for pennation angle of VL (*p* = 0.169; ES = 0.583; CI: −0.30, 1.64).

Results for performance parameters of male and female groups are reported in Table 2. ANCOVA revealed a significant difference after adjusting for LBM between group means for bench press 1RM (F = 6.224; *p* = 0.019; η^2^ = 0.187; −59.3% in women compared to men); BPT (F = 6.706; *p* = 0.015; η^2^ = 0.199; −61.5% in women compared to men). No significant differences between male and female groups after adjusting for LBM were detected for squat 1RM (F = 0.005; *p* = 0.945; η^2^ = 0.001); deadlift 1RM (F = 0.531; *p* = 0.472; η^2^ = 0.019) and CMJP (F = 0.202; *p* = 0.656; η^2^ = 0.007). When adjusted for body mass, all performance variables were significantly different between male and female groups (*p* < 0.05; see Table 2). After adjusting for the muscle thickness of PEC, ANCOVA revealed a significant difference between genders for bench press 1RM (F = 4.798; *p* = 0.037; η^2^ = 0.151) and for bench press throw power (BPT) (F = 4.669; *p* = 0.040; η^2^ = 0.147). Significant differences after adjusting for VLMT were detected for squat 1RM (F = 14.002; *p* = 0.001; η^2^ = 0.341), 1RMDE (F = 20.061; *p* < 0.001; η^2^ = 0.426), MSP (F = 17.776; *p* < 0.001; η^2^ = 0.397), and CMJP (F = 22.012; *p* < 0.001; η^2^ = 0.449).

All correlations between strength and power performances and muscle architecture in the male group are reported in Table 3. Very high correlations were observed in men between PECMT and bench press 1RM (*r =* 0.83; *p* < 0.001; CI: 0.56, 0.93); and between PECMT and BPT (*r =* 0.89; *p* < 0.001). Very high correlations were also found in the same group between mid-shin pull (MSP) and squat 1RM, deadlift 1RM, and CMJP (*r =* 0.66–0.90; *p* < 0.01). In addition, lean body mass showed very high correlations with squat 1RM (*r =* 0.80; *p* < 0.001; CI: 0.22, 0.86), deadlift 1RM (*r =* 0.78; *p* < 0.001; CI: 0.22, 0.86), bench press 1RM (*r =* 0.70; *p* = 0.003; CI: 0.22, 0.86), CMJP (*r =* 0.91; *p* < 0.001; CI: 0.22, 0.86), MSP (*r =* 0.88; *p* < 0.001; CI: 0.22, 0.86). A moderate correlation was observed between lean body mass and the muscle thickness of PEC (*r =* 0.53; *p* = 0.033; CI: 0.22, 0.86).

Correlations between strength and power performances and muscle architecture in the female group are reported in Table 4. Very high correlations were observed in women between the muscle thickness of PEC and bench press 1RM (*r =* 0.82; *p* < 0.001; CI: 0.51, 0.94); and between the muscle thickness of PEC and bench press throw power (*r =* 0.82; *p* < 0.001; CI: 0.52, 0.91). In addition, lean body mass showed high correlations with squat 1RM (*r =* 0.60; *p* = 0.024; CI: 0.10, 0.86), bench press 1RM (*r =* 0.67; *p* = 0.008; CI: 0.21, 0.87), CMJP (*r =* 0.68; *p* = 0.007; CI: 0.23, 0.89), and MSP (*r =* 0.80; *p* < 0.001; CI: 0.46, 0.93). A high correlation was observed between lean body mass and the muscle thickness of PEC (*r =* 0.74; *p* = 0.002; CI: 0.34, 0.91).

## 4. Discussion

The aim of the present investigation was to compare male and female resistance trained athletes in absolute and relative strength and power performances. Firstly, women had lower maximal strength values when compared to men at bench press (−59.2%), squat (−57.2%), deadlift (−56.3%), and mid-shin pull (MSP, −53.2%). In addition, lower levels of power were detected in females in both the upper (−61.2%) and the lower body (−44.2%). This is consistent with previous studies [5,6] that reported similar differences between men and women in the upper body. The same authors however, reported that women were only 27% less strong than men in lower body strength. The larger differences found in the present investigation between male and female athletes may be related to the strength assessments performed. Some of these maximum strength assessments (e.g., deadlift 1RM), are deeply influenced by the upper-body strength [34]. These findings indeed, are similar to those previously reported for powerlifters of both sexes [35,36,37].

The results of the present investigation partially confirmed the research hypothesis. However, no differences between sexes were found in lower body maximal strength and power adjusted for lean body mass (LBM). On the contrary, higher maximal strength and power adjusted for LBM were detected in the upper body. Some authors suggested that differences in the upper body strength performance may be related to a different muscle mass distribution between males and females rather than to a different neuromuscular function [1]. In the present study, however, differences between genders in maximal strength and power were still significant when adjusting for the thickness of the main muscles involved in each exercise. A significant difference in squat and deadlift 1RM adjusted for muscle thickness of vastus lateralis, without a significant difference when adjusted for LBM, may support the idea that man and women used different movement strategies during weight-bearing exercises [38]. A significant difference between sexes in anaerobic power, regardless of LBM, has been previously reported by Mayhew et al. [7] and Perez-Gomez et al. [15]. Differences between sexes in power were accounted for muscle fiber types, muscle quality, or glycolitic enzymatic activities [39].

Regarding muscle thickness, present findings indicate that male individuals were characterized by significantly higher values of this parameter in trapezius, pectoral, and vastus lateralis muscles, compared to females. In addition, fascicle length of vastus lateralis, were significantly longer in males compared to females. Muscle fascicle length (FL) has been suggested to play an important role in determining the maximum contraction velocity of the muscle [20,40] and the range of active force production. Longer fascicles indeed, warrant higher contraction velocity through a great number of sarcomeres in series [17,20] and may be an advantage when strength is produced from a deep squatting position [16,41]. In addition, previous investigations have reported strong correlations between FL of the vastus lateralis and squat and deadlift 1RMs, in competitive powerlifters [35]. Higher FL measured in males compared to females, may partially explain the difference in strength and power, relative to muscle thickness, detected in the present study. Differences in this parameter of muscle morphology, may not be related to the individual resistance training experience since both groups participating in the present investigation had similar experience in resistance training and were involved in high demanding strength workouts for at least three years. No significant differences between the sexes were detected in the pennation angle of the vastus lateralis (VLPA); values of VLPA however, were lower compared to the values measured in competitive powerlifters [35]. Heavy resistance training has been associated with increases in the pennation angle. On the contrary, no changes in this parameter were detected when high speed power training was included in the resistance training program [19]. The participants in the present study were involved in both resistance and high-speed power training and the combined effects of these training contents may have limited the increase in pennation angle.

As suggested by previous investigations [35], lean body mass (LBM) can be considered as one of the most important factors for maximal strength and power performance and, in our study, this parameter was significantly correlated with the fascicle length of the vastus lateralis (VLFL). This is consistent with Kearns et al. [42] and Brechue et al. [35] that demonstrated an important role of fascicle length for muscle mass accumulation. High fascicle lengths may possibly represent either a genetic predisposition for muscle mass accumulation or a specific adaptation to resistance training consisting in fascicle lengthening and radial hypertrophy [17,43].

High correlations have been noted in both male and female athletes between the different strength exercises performed in the present study. Very high correlations have been also detected between the mid-shin pull test (MSP) and the squat, deadlift, and bench press 1RMs. Despite the different lifting mechanics that characterize these exercises [44], large relationships confirm that MSP may represent a synthetic parameter encompassing the individual maximal strength capabilities [24]. Curiously, the correlation between MSP and the countermovement jump power (CMJP) was higher in men (*r* = 0.85) compared to women (*r* = 0.52). In addition, significant correlations between the fascicle length of vastus lateralis and the squat and deadlift 1RMs, were found in men only. This difference may be due to different movement strategies between the sexes [38] that may influence the contribution of the different muscle groups to the final performance.

In conclusion, the results of the present investigation indicate that significant differences in strength and power relative to body mass, lean body mass, and muscle thickness exist between male and female strength and power athletes. The present study also confirms that deep relationships exist between strength and power performance and muscle architecture in trained athletes. In particular, high correlations exist in both sexes between the muscle thickness of pectoral muscles and both bench press 1RM and power. A limitation of the present investigation is that lifting technique and muscle activation were not assessed and differences between men and women cannot be accounted for lifting strategies. Another limitation may consist in the use of a single muscle thickness to adjust the 1RM of multi-joint exercises. In addition, strength and conditioning programs followed by the participants in the years preceding this study, likely differ in several variables (e.g., number of upper and lower body exercises, training frequency, periodization model, etc.), and may have deeply influenced both performance and muscle morphology.

## Figures and Tables

**Figure 1 jfmk-06-00017-f001:**
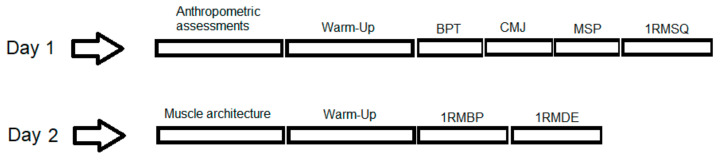
Timeline of the assessments performed in Day 1 and Day 2. MSP = Mid Shin Pull test; BPT = bench press throw; CMJ = countermovement jump; 1 repetition maximum (1RM)SQ = 1RM squat; 1RMBP = 1RM bench press; 1RMDE = 1RM deadlift.

**Table 1 jfmk-06-00017-t001:** Mean ± SD anthropometric and muscle architecture values in men and women. FM = fat mass; LBM = lean body mass; TRAPMT = muscle thickness of trapezius muscle; PECMT = muscle thickness of pectoralis muscle; VLMT = muscle thickness of vastus lateralis; VLPA = pennation angle of vastus lateralis; VLFL = fascicle length of vastus lateralis. * denotes a significant (*p* ≤ 0.05) differences between groups.

	Men	Women
FM (kg)	13.1 ± 3.6	11.3 ± 2.0
LBM (kg)	75.8 ± 12.6	44.5 ± 4.3 *
TRAPMT (mm)	13.9 ± 2.7	8.8 ± 1.9 *
PECMT (mm)	22.9 ± 3.2	12.5 ± 3.3 *
VLMT (mm)	20.5 ± 3.6	14.9 ± 3.2 *
VLPA (°)	8.5 ± 1.2	7.8 ± 1.2
VLFL (mm)	14.5 ± 2.7	10.7 ± 2.3 *

**Table 2 jfmk-06-00017-t002:** Mean ± SD values for maximal strength and power in men and women. SQ = squat; DE = deadlift; BP = bench press; MSP = mid-shin pull; BPT = bench press throw; CMJP = countermovement jump power. F = ANCOVA F η^2^ = partial eta squared. ANCOVA = analysis of covariance.

	Men	Women	ANCOVA Statistics (Difference Adjusted for BM)	ANCOVA Statistics (Difference Adjusted for LBM)
1RMSQ (kg)	178.5 ± 50.3	76.4 ± 31.3	F = 29.731	F = 0.005
			*p* < 0.001	*p* = 0.945
			η^2^ = 0.219	η^2^ = 0.001
1RMDE (kg)	203.4 ± 49.5	88.7 ± 32.4	F = 12.339	F = 0.531
			*p* = 0.002	*p* = 0.472
			η^2^ = 0.314	η^2^ = 0.019
1RMBP (kg)	121.1 ± 23.4	49.3 ± 16.6	F = 29.731	F = 6.224
			*p* < 0.001	*p* = 0.019
			η^2^ = 0.524	η^2^ = 0.187
MSP (N)	2030.5 ± 545.1	943.2 ± 210.8	F = 9.613	F = 0.462
			*p* = 0.004	*p* = 0.502
			η^2^ = 0.263	η^2^ = 0.017
BPT (W)	477.7 ± 115.2	185.4 ± 55.7	F = 23.643	F = 6.706
			*p* < 0.001	*p* = 0.015
			η^2^ = 0.467	η^2^ = 0.199
CMJP (w)	4613.9 ± 925.6	2577.6 ± 547.0	F = 10.856	F = 0.202
			*p* = 0.003	*p* = 0.556
			η^2^ = 0.287	η^2^ = 0.007

**Table 3 jfmk-06-00017-t003:** Correlations between the different parameters of performance and muscle morphology in male group. 1RMSQ = 1RM squat; DL = deadlift; BP = bench press; MSP = mid-shin pull; BPT = bench press throw; CMJP = countermovement jump power; TRAPMT = muscle thickness of trapezius muscle; PECMT = muscle thickness of pectoral muscle; VLMT = muscle thickness of vastus lateralis.

	1RMSQ	1RMDL	1RMBP	MSP	POW50	CMJP	TRAPMT	PECMT	VLMT
1RMSQ		0.95; *p* < 0.001	0.80; *p* < 0.001	0.90; *p* < 0.001	0.38; *p* = 0.144	0.75; *p* < 0.001	0.66; *p* = 0.005	0.44; *p* < 0.085	0.65; *p* = 0.006
1RMDL			0.85; *p* < 0.001	0.86; *p* < 0.001	0.43; *p* = 0.09	0.78; *p* < 0.001	0.61; *p* = 0.013	0.55; *p* = 0.027	0.55; *p* = 0.028
1RMBP				0.66; *p* = 0.006	0.54; *p* = 0.032	0.76; *p* < 0.001	0.71; *p* = 0.002	0.83; *p* < 0.001	0.41; *p* = 0.112
MSP					0.55; *p* < 0.027	0.85; *p* < 0.001	0.48; *p* < 0.056	0.49; *p* < 0.097	0.55; *p* = 0.029
BPT						0.67; *p* = 0.004	0.46; *p* < 0.070	0.57; *p* < 0.022	0.03; *p* = 0.917
CMJP							0.37; *p* = 0.156	0.64; *p* = 0.008	0.36; *p* < 0.170
TRAPMT								0.45; *p* = 0.079	0.07; *p* = 0.483
PECMT									0.19; *p* = 0.485

**Table 4 jfmk-06-00017-t004:** Correlations between the different parameters of performance and muscle morphology in female group. 1RMSQ = 1RM squat; DL = deadlift; BP = bench press; MSP = mid-shin pull; BPT = bench press throw; CMJP = countermovement jump power; TRAPMT = muscle thickness of trapezius muscle; PECMT = muscle thickness of pectoral muscle; VLMT = muscle thickness of vastus lateralis.

	1RMSQ	1RMDL	1RMBP	MSP	POW50	CMJP	TRAPMT	PECMT	VLMT
1RMSQ		0.94; *p* < 0.001	0.93; *p* < 0.001	0.75; *p* = 0.02	0.68; *p* = 0.007	0.31; *p* = 0.28	0.52; *p* = 0.056	0.72; *p* = 0.004	0.19; *p* = 0.518
1RMDL			0.84; *p* < 0.001	0.69; *p* = 0.009	0.66; *p* = 0.010	0.28; *p* = 0.329	0.4; *p* = 0.159	0.61; *p* = 0.019	0.28; *p* = 0.335
1RMBP				0.77; *p* = 0.001	0.66; *p* = 0.010	0.47; *p* < 0.088	0.71; *p* = 0.005	0.82; *p* < 0.005	0.26; *p* = 0.373
MSP					0.66; *p* < 0.035	0.52; *p* = 0.050	0.62; *p* = 0.018	0.72; *p* = 0.003	0.19; *p* = 0.502
BPT						0.43; *p* = 0.122	0.41; *p* < 0.143	0.82; *p* < 0.001	0.25; *p* = 0.380
CMJP							0.42; *p* = 0.130	0.57; *p* = 0.034	0.06; *p* = 0.845
TRAPMT								0.70; *p* = 0.006	0.33; *p* = 0.242
PECMT									0.32; *p* = 0.257

## Data Availability

Data available on request due to restrictions (privacy).

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
