# Peer review of "A Comparison between Male and Female Athletes in Relative Strength and Power Performances"

_jfmk, 2021, doi:10.3390/jfmk6010017_

Round 1

Reviewer 1 Report

Please refer to the attached document. 

Author Response

Rew 1

As stated in lines 8-9, ………

The authors appreciated the reviewer’s suggestions and comments. The manuscript has been deeply revised as suggested by the reviewer.

Specific Comments Abstract Section

Please clarify the study aims. Particularly because you report two different ones (please see introduction and abstract).

Revise your conclusion according to study findings.

Line 9. One word, no need to abbreviate. Same in W. In line 10, years, kg, cm... add space after values.

Amended

Line 16. Please use male and female. Revise throughout. Line 17, Please change to gender differences, and modify throughout. Sex is not the correct word to use in this context.

Amended

Line 18. Abbreviations needs to be explained on 1st appearance, please modify in all cases (i.e. 1RMDE, etc.). Revise throughout.

Amended

Introduction

Lines 26-28. This is confusing, and in turn could be reduced to one only and more clear sentence. Please reword.

Amended. The periods have been changed as follows: “Muscle strength of women indeed, is typically reported in the range of 40 to 75% of that of men [2] and women are also known to be less powerful than equally trained men. [3].”

Lines 29-30. Gender differences are still evident when relative power values are considered...Please rephrase. In line 30, do authors refer to absolute or relative values here?

The subsequent period: “lower absolute levels of muscle power expressed in weightlifting exercises have been reported in women compared to men” has been replaced with: “and women are also known to be less powerful than equally trained men.

Line 34. No need to use ‘sex’ here. Please delete.

Amended

 Line 35. Please justify this assumption or delete.

The following words have been removed: “and body mass distribution in the upper and lower body”

Lines 36-38. Please revise. Strength, power and anaerobic power are all neuromuscular factors associated to performance.

The period has been changed as follows: “Other studies confirmed that gender differences in strength may be accounted to LBM but reported that the differences in power performances were still apparent regardless of body composition, and muscle mass [10,11], supporting the idea that differences between genders in anaerobic power and jumping capacity could not be accounted for by differences in lean body mass only [10,11].

Lines 38-40. Same as previous, maybe both sentences could be merged and reword into a clearer statement.

Amended, please see the previous point.

Lines 41-42. Despite no significant differences were found between genders in muscle fibres numbers... How do authors explain 'qualitative difference' in muscle tissue? Please clarify and include a reference.

The periods have been changed as follows: “Despite no significant differences between genders in muscle fibers number were reported [6], a qualitative difference in muscle tissue, such as a higher concentration of glycolytic enzymes and greater proportion of fast type muscle fibers [12], may explain the disparity in strength.”

Line 43. Please try to connect the previous sentence to this new one. As it stands, the sentence appears out of the blue.

Amended, please see the previous point.

Lines 45. Do authors mean FT fibers CSA? Please clarify. Line 46. Please delete 'between sexes'…

Amended “…as the muscle area occupied by fast type fibers…”

Line 48. More recently, however, ... revise.

Lines 49-51. Not only this factors affect strength and power performance. Please expand and clarify. Lines 51-52. Please try to connect both sentences... Please consider.

Line 48; 49-51; 51-52. The periods have been changed as follows: “Maximal strength and power are influenced by many neuromuscular factors including muscle morphological characteristics such as muscle thickness, pennation angle and fascicle length [16,17]. Recently, moderate correlations have been reported between maximal isometric force expressed at the mid shin pull and muscle architecture of the vastus lateralis in resistance-trained individuals [16]. These parameters of muscle may be influenced by resistance training architecture in both male and female athletes [18,19].”

Lines 53-55. This sentence doesn’t add information to clarify the study aims/problem of the present. Delete or further expand to provide a clearer picture with regards to what you are trying to demonstrate.

The period has been moved earlier to follow the section flow.

Line 57. performances of female and male athletes in relation to... Please reword.

Amended. The period has been changed as follows: “To the best of our knowledge, no studies to date have compared strength and power performances of male and female athletes in relation to body composition and muscle architecture.”

Lines 58-60. Please consider to use the aim stated in the abstract. 'The aim of this study was to compare male vs. female athletes in strength and power performance relative to lean body mass (LBM) and muscle architecture'. Authors also needs to be absolutely clear regarding study aims: in the abstract it states performance measure relative to LBM and muscle architecture. Whereas here it states adjusted to body mas, LBM and muscle architecture. Finally, in table 2, there is only two ANCOVAS one for BM and the other adjusted to LBM.

As suggested by the reviewer, the following period: “Thus, the aim of the present investigation was to compare male and female strength and power athletes in maximal strength and power relative to body mass, lean body mass and muscle architecture.” Has been replaced with: “The aim of this study was to compare male vs. female athletes in strength and power performance relative to body mass, lean body mass and muscle architecture.”

Line 60. A second aim was to... please reword.

Amended:” A second aim of the present study was to assess the relationships between muscle architecture and maximal strength in resistance trained athletes competing in sports with elevated strength components.”

Line 63. Reword size to thickness. Regarding the hypothesis, it will add more value if authors can include references to justify this... Authors also must confirm whether adjusted BM values will be included or not.

Amended. Body mass has been included. References 10 and 11 have been added.

Methods Section

Please include the study design: cross-sectional, descriptive and correlational.

The following period has been included in the manuscript: “In the present cross-sectional, descriptive and correlational study, participants reported to the laboratory on two separate occasions….”

Line 66 Can authors explain how participants were included?

  The following words have been included in the Participants section: “All the participants volunteered to take part in the present study,”

Line 76. Exclusion criteria and line 78 are repeated.. Please delete.

Amended

Line 82. One major concern arise here…….

Despite the authors believe that normalization of athlete performances may facilitate the comparison between different populations, in the present investigations body mass, lean body mass or muscle thickness are used as covariates. Thus, the authors believe that further normalizations are not necessary.

Line 84-86 This is not what is shown in figure 1….

Amended. The period has been changes as follows. “for upper and lower body power and for maximal isometric strength…”

Line 95: this was actually prformed at the beginning of the session, not before. The following sentences could be merged and simplified…Please consider.

Amended. The period has been changed and merged with the following: “Anthropometric evaluations were performed at the beginning of the first assessment session, and included body mass, height and body fat composition.”

Line 98-99. Amended

Line 99. Amended

Line 111 Mid Shin Pull (MSP)test represents an isometric assessment, thus there is no power (mean or peak) expression. The authors agree that the RFD expressed at MSP may represent an important parameter, however, due to the high variability measured in previous studies in our lab, this parameter was not considered.

Lines 111-112  Absolute and relative reliability must be reported……

CCs and CVs were included in the results sections and explained in the Statistical Analysis section.

The following period has been included in the results section: “Intraclass coefficients of correlation were 0.94 (SEM = 158.4 N; CV = 0.22), 0.99 (SEM = 32.56; CV = 0.19) and 0.96 (SEM = 100.3 w; CV = 0.20) for PF at MSP, BPT and CMJP, respectively.” In addition, these words were included in the methods: “A relative reliability index (intra-class correlation coefficient, ICC) was used to examine the level of agreement between the attempts performed in each assessment. Absolute reliability (standard error of measurement, SEM) was used to define the extent to which a score varies on test-retest measurements. Coefficients of variation were calculated using the individual best performance in each assessment.”

Line 113-114 Lines 113-114. Did authors perform CMJ at the end of the session? Jumps are very sensitive to fatigue;so don’t you think this could have been affected by the previous exercises performed? What was the reliability? What about 1RMSQ? if this was done on the first session it must be explained here, not in

the next paragraph. Please revise.

The authors agree that jumps are very sensitive to fatigue. As reported in figure 1 indeed, CMJ was not performed at the end of the training session, but at the beginning of Day 1, following the bench press throw test.  The order of the strength and power assessments described in paragraph 2.3 has been updated. In addition, the description of the bench press throw test has been updated as follows: “Participants pressed loads corresponding to 50% of their estimated 1RM (based on the loads used in training). Two sets of 2 repetitions were required with a recovery time of 20 s between repetitions and of 3 min between sets.” Further information have been included regarding CMJ: “They were asked to perform 3 jumps with a recovery time of 3 minutes between the attempts.” In addition, CMJ absolute and relative reliability were included: 0.96 (SEM = 100.3 w).

The order of the exercises within the methods has been updated.

Lines 125-131: Lines 125-131. This test should be explained in the paragraph where authors describe tests of session1, and in the exact same order as they mentioned and explained in the abstract and figure 1. Second, an optical encoder doesn’t measure mean power neither peak, but instead it measures mean velocity. Power

values derived from this test shouldn’t be reported. Please delete. You may wish to include mean

velocity, which is fine. Authors are also asked to include a reference to justify the use of this technology

(validation).

The authors agree that linear encoders measure the velocity of the bar, however,  mean power can be easily calculated knowing the load lifted. The period has been changed as follows: “an optical encoder (Tendo Unit model V104; Tendo Sports Machines, Trencin, Slovak Republic) was used to measure the mean velocity and to calculate the power generated. “

Several scientific investigations have used this technology to calculate mean power expressed in linear resistance exercises:

Hoffman, J. R., Ratamess, N. A., Kang, J., Rashti, S. L., & Faigenbaum, A. D. (2009). Effect of betaine supplementation on power performance and fatigue. Journal of the International Society of Sports Nutrition6(1), 7.

Pennington, J., Laubach, L., De Marco, G., & Linderman, J. (2010). Determining the Optimal Load for Maximal Power Output for the Power Clean and Snatch in Collegiate Male Football Players. Journal of Exercise Physiology Online13(2).

Line 132. Verbally encouraged to do what exactly?

Line 132.  The period has been changed as follows: “During all strength measurements, participants were verbally encouraged by the study investigators to exert maximum effort in each attempt.”

Lines 138-139. Any reasoning as to why authors selected this muscles? Please justify.

The following period has been included in the section: “These muscles were selected because previous studies indicated a good reliability of the measurements in resistance trained individuals (Bartolomei, S., Nigro, F., Lanzoni, M., Masina, F., Di Michele, R., & Hoffman, J. R. (2020). A Comparison Between Total Body and Split Routine Resistance Training Programs in Trained Men. Journal of Strength and Conditioning Research.).”

Line 150. This is not what was reported in the methods section and figure 1. Revise.

Amended.

Lines 157-160. Please include a reference here

Lines 157-160 A reference has been included  : (26) Bartolomei, S., Sadres, E., Church, D. D., Arroyo, E., Gordon III, J. A., Varanoske, A. N., ... & Hoffman, J. R. (2017). Comparison of the recovery response from high-intensity and high-volume resistance exercise in trained men. European journal of applied physiology117(7), 1287-1298.

Statistical Analysis:

Please include details with regards to absolute and relative reliability. Second, when performing a between groups comparison, in this case an independent t test should be used. In addition to this tests, effect sizes and confidence intervals should be included to provide a clearer interpretation of results. Harrison, A. J., McErlain-Naylor, S. A., Bradshaw, E. J., Dai, B., Nunome, H., Hughes, G. T., ... & Fong, D. T. (2020). Recommendations for statistical analysis involving null hypothesis significance testing.

Details about absolute and relative reliability have been included.

Anova has been replaced with independent t tests. ES (Hedge’s g) and CI values have been reported. The following period has been included in the statistical analysis section: “Independent sample t tests were used to compare the mean values of body composition and muscle architecture variables between sexes. In addition, Edge’s g effect size (ES) and 95% confidence intervals (CI) were reported.”

Lines 173-174. When performing correlations analysis, with such a heterogeneous n (male & female) the outcome is highly probable that you may find spurious correlations. Which is obvious as shown in table 3, all correlations are significant and high to nearly perfect. Do they really make sense? This must be performed separately, no matter a lower n. This another major concern derived from my analysis. The correlations coefficients must be reported combined with their respective confidence intervals.

Correlations are now presented for men and women in  different tables. In addition, confidence intervals are reported in the results.

Results section

Lines 191-194. This must be reported below tables.

Amended

Line 202. Adjusted to PECMT? This must be included in table 2, and VLMT and TRAP MT also reported

The authors understand the reviewer’s query but significant differences are already reported in the results. In addition, adjusting lower body strength parameters (e.g., 1RM squat) for the muscle thickness of upper body muscles (e.g., PectoralMT), may not be so interesting.

Lines 207-209. This must be reported below table 2, and should include what does F and n2 refers to.

Line 207-209. Amended

Line 210. Please delete 3.1. and start the sentence as it is in line 211

Line 210.  Amended

Lines 222-224. Please report below table 3. Table 3. Delete number 1 in all cases.

Amended. Table 4 has been included.

Discussion Section

 Lines 227-228. The aim needs to be rephrased, it is not clear as it stands. Rugby players are team-sports athletes not strength and power athletes. Please reword.

Amended: The period has been changed as follows: “The aim of the present investigation was to compare male and female resistance trained athletes in absolute and relative strength and power performances.”

Lines 228-229. Please reword. This needs to be clear and supported by findings. Are this differences in absolute and/or relative values and regarding what measures?

Amended, see lines 227-229

Lines 227-229.  The periods have been changed as follows: “Firstly, W were weaker than M in maximal strength expressed at bench press (-59.2%), squat (--57.2%), deadlift (-56.3%) and mid-shin pull (MSP, -53.2%). In addition, lower levels of power were detected in females in both the upper (-61.2%) and the lower body (-44.2%).”

 Lines 233-234. How do authors claim that SQ and DE are deeply influenced by upper body strength? Based on what grounds?

The period has been changed as follows: “Some of these maximum strength assessments (e.g. deadlift 1RM), are deeply influenced by the upper-body strength (Bezerra ES, Simão R, Fleck SJ, et al. Electromyographic activity of lower body muscles during the deadlift and still‒legged deadlift. Journal of Exercise Physiology Online. 2013;16(3):30‒39)”

Lines 236-252. This paragraph shows little to no discussion around the differences observed and doesn’t

clarify reasons.

Discussion has been revised. Please see the following paragraphs

Line 244. I would suggest authors to refer to the specific test instead to maximal strength as this is

something is not measured during 1RM tests.

The period has been changed as follows: “A significant difference in squat and deadlift 1RM adjusted for MT of vastus lateralis, without a significant difference when adjusted for LBM, may support the idea that men and women used different movement strategies during weight-bearing exercises [30].”

Lines 243-245. This is probably due to the fact than more than one muscle groups are activated in

exercises, so maybe to account for MT in only one muscle is not representative of relative strength in a

specific action.

Another limitation, about the use of a single muscle MT to adjust a multi-joint exercise 1RM, has been included at the end of the section.

Lines 246-247. Limitations must be reported in the last paragraph.

Amended. In addition the following changes have been done: “A limitation of the present investigation is that lifting technique and muscle activation were not assessed and differences between men and women cannot be accounted for lifting strategies. Another limitation may consist in the use of a single muscle thickness to adjust the 1RM of multi-joint exercises”.

Lines 251-252. This is speculative as authors are not in a position to state such a sentence.

Amended. The period has been removed.

 Line 253. Start a new paragraph: Regarding muscle thickness, present findings indicate that...

Amended. The period has been changed as follows: “Regarding muscle thickness, present findings indicate that male individuals were characterized by significantly higher values of this parameter in trapezius, pectoral and vastus lateralis muscles, compared to females.”

Line 261. Please reword to highly trained athletes and clarify if you refer to weightlifting or other.

The period has been changed as follows. “In addition, previous investigations have reported strong correlations between VLFL and squat and deadlift 1RMs, in competitive powerlifters [27].”

Lines 267-269. This is an important limitation and must be reported in the specific section.

The following period has been included at the end of the section:”In addition, strength and conditioning programs followed by the participants in the years preceding this study, likely differ   in several variables (e.g., number of upper and lower body exercises, training frequency, periodization model etc.), and may have deeply influenced both performance and muscle morphology.”

Lines 277-288. I will not further comment regarding correlations as this must be done for males and

females in a separate analysis.

The comments regarding the correlations have been changed as follows: “High correlations have been noted in both male and female athletes between the different strength exercises performed in the present study. Very high correlations have been also detected between the mid-shin pull test (MSP) and the 1RM in squat, deadlift and bench press. Despite the different lifting mechanics that characterize these exercises [36], large relationships confirm that MSP may represent a synthetic parameter encompassing the individual maximal strength capabilities [22]. Curiously, the correlation between MSP and the countermovement jump power (CMJP) was higher in men compared to women. In addition, significant correlations between the fascicle length of vastus lateralis and the squat and deadlift 1RMs, were found in men only. This difference may be due to different movement strategies between the sexes [30] that may influence the contribution of the different muscle groups to the final performance.”

Lines 289-294. Conclusions must clear, brief and supported by study findings.

The following period has been removed from the conclusions “Superior lower body strength and power performances relative to muscle thickness in M may be related to the longer fascicle length detected in this group compared to W.” The paragraph has been changed as follows: “In conclusion, the results of the present investigation indicate that significant differences in strength and power relative to body mass, lean body mass, and muscle thickness exist between male and female strength and power athletes. The present study also confirms that deep relationships exist between strength and power performance and muscle architecture in trained athletes. In particular, high correlations exist in both sexes between the muscle thickness of pectoral muscles and both bench press 1RM and power. “

Reviewer 2 Report

Thank you to the authors for your work on this manuscript. I believe that this study is interesting and fills the gap in sport science literature. I feel that overall the design and preparation of the manuscript was well done but it will be useful to add possibilities of practical applications of these results.

43: explain abbreviations
48: please, rewrite this sentence

Do differences in hormones have an impact on the strength and power performances of athletes of both sexes?
what performance and phase of the menstrual cycle? please add this information to the Introduction section.

66: How was the sample size determined? Was the study adequately powered?

70, 71: use abbreviations

70- 72; please provide information separately for W and M

78-79: It was mentioned above
81: Did you control the menstrual cycle? in which phase was the women?
80:What was with other supplements? It was allowed? What about training routines and dietary patterns during the study? What was it controlled? Did participants report diet 24 h before too?

81: provide a number

99: What kind of calliper did you use?

113- how many jumps did they perform?
116- “Bench press 1RM, squat 1RM and deadlift 1RM tests were
performed as described in previous studies [16]”. It was only one ref.
Further this ref. [16] applies only to deadlift (not directly, only
later to Hoffman 2014 so for what is ref Bartolomei et al., 2019) and
not to BP and SQ. For better transparency, please describe the
procedures of testing used in this manuscript.
According to PMID: 32390725; PMID: 32269656 during the 1RM test and
experimental trials the tempo of movement should be defined. Please add
such information and refs. However, if it was not controlled, please
write that the movement tempo was volitional according to these refs.
Please add the width grip used during the bench press according to PMID:
31531132
Was a test-retest of 1-RM carried out? The authors may provide the
variation coeficient?
What is the reliability of the used tests? Of for tendo

Tables- The description of tables are not appropriate
Table 1: I believe that information about % of BF will be useful
Table 2:
change w to W
BBT and CMJP- which repetition (s?) was/were included in the analysis?

131: I don't understand- on different loads?
Table 2: in case of power- provide information about mean and peak power

218- FFM or LBM?

229- in relative or maximal strength?

226- I believe the first paragraph has to consist only information about the results of this study
251- 252- in this study or generally? please provide reference

264- 267- provide more specific information to section participants

You have to provide all limitations, idea for future research and practical applications at the end of the study.

Author Response

Rew 2

The authors would like to thanks the Reviewer 2 for his comments and suggestion to improve the quality of the manuscript

43: Explain abbreviations

Amended

48: please rewrite this sentence

The period has been changed as follows: More recently however, Perez-Gomez et al. [15], found similar power output in both sexes, when results were normalized to the muscle mass of the lower limbs.”

Do differences in hormones have an impact on the strength and power performances of athletes of both sexes?
what performance and phase of the menstrual cycle? please add this information to the Introduction section.

The authors agree with the reviewer that the phase of the menstrual cycle may represent an important variable. However, it was not controlled in the present investigation. In addition, a recent systematic review concluded that the phase of the menstrual cycle has minimal influences on strength performance (Blagrove, R. C., Bruinvels, G., & Pedlar, C. R. (2020). Variations in strength-related measures during the menstrual cycle in eumenorrheic women: A systematic review and meta-analysis. Journal of Science and Medicine in Sport.)

  1. How was the sample size determined? Was the study adequately powered?

The following period has been included in the methods: “The estimated sample size was 20 to detect a between-group difference of 200 w and of 30 kg in upper body power and strength, respectively.”

70, 71: use abbreviations

 As requested by reviewers 1 and 3, males and females (women) were not abbreviated.

70-72: Information for males and females have been provided.

78-79: it was mentioned above

Amended

81:   The phase of the menstrual cycle was not controlled. Please see previous comments.

  1. What was with supplements…..

The use of supplements was not allowed  in the 24h before the study and the following words have been included in the section: “Participants were asked to abstain from alcohol, caffeine, any dietary supplement, and resistance training for at least 24 h prior to the tests.” however, the diet was not controlled. About the training program, the following limitation has been included: “In addition, strength and conditioning programs followed by the participants in the years preceding this study, likely differ in several variables (e.g., number of upper and lower body exercises, training frequency, periodization model etc.), and may have deeply influenced both performance and muscle morphology.”

  1. Provide a number

The authors did not understand the reviewer’s query

  1. What kind of calliper did you use?

The following information have been included: “..using an Harpender Skinfold Caliper (Harpenden, British Indicators, West Sussex, UK).”

  1. How many jumps did they perform?

The following period has been included: They were asked to perform 3 jumps with a recovery time of 3 minutes between the attempts.“

  1. Bench press 1RM, squat 1RM and deadlift 1RM tests were performed as described in previous studies [16]”. It was only one ref. Further this ref. [16] applies only to deadlift (not directly, only
    later to Hoffman 2014 so for what is ref Bartolomei et al., 2019) and not to BP and SQ. For better transparency, please describe the procedures of testing used in this manuscript.

The following additional information have been included in the section to better describe the 1RM assessments: “The deadlift 1RM assessment was performed using an Olympic bar and plates (Pallini Sport Inc. Malaquis, France). The same barbell and plates were used for squat and bench press 1RMs. During the squat 1RM, participants were asked to unrack the barbell and squat from a standing position, until the greater trochanter of the femur was at the same level of the knee”  ; “Then, participants pushed the bar until his arms were fully extended. Their grip widths were recorded to reproduce the same hands position in all the attempts.”

According to PMID: 32390725; PMID: 32269656 during the 1RM test and experimental trials the tempo of movement should be defined. Please add such information and refs. However, if it was not controlled, please write that the movement tempo was volitional according to these refs. Please add the width grip used during the bench press according to PMID:31531132

Amended. The following period and reference has been included in the section: “” Wilk, M., Golas, A., Zmijewski, P., Krzysztofik, M., Filip, A., Del Coso, J., & Tufano, J. J. (2020). The effects of the movement tempo on the one-repetition maximum bench press results. Journal of human kinetics72(1), 151-159.

Was a test-retest of 1-RM carried out? The authors may provide the variation coeficient?
What is the reliability of the used tests? Of for tendo

ICC s and  SEM were reported for all the assessments performed. However, no test-retests were performed for 1RM assessments.

Tables- The description of tables are not appropriate

Descriptions have been checked.Table 1: I believe that information about % of BF will be useful

The investigation takes into account several parameters and the authors believe that using both %FM and FM in kg may be redundant.Table 2:change w to W

AmendedBBT and CMJP- which repetition (s?) was/were included in the analysis?

The following period has been included in the methods: “In both CMJP and BPT assesments, the best trials were registered and used for analysis.

131: I don't understand- on different loads?

..on different loads has been removed. It was a typo.Table 2: in case of power- provide information about mean and peak power

The authors have calculated both mean and peak power. However ANCOVA and correlational analysis did not provide any additional information than mean power. Then, the authors decided to remove this parameter in order to increase the clarity of the paper.

218- FFM or LBM?

Amended. FFM was replaced with LBM

229- in relative or maximal strength?

The period has been changed as follows: “The aim of the present investigation was to compare male and female resistance trained athletes in absolute and relative strength and power performances”226- I believe the first paragraph has to consist only information about the results of this study

The paragraph has been changed as follows: “The aim of the present investigation was to compare male and female resistance trained athletes in absolute and relative strength and power performances. Firstly, W were weaker than M in maximal strength expressed at bench press (-59.2%), squat (--57.2%), deadlift (-56.3%) and mid-shin pull (MSP, -53.2%). In addition, lower levels of power were detected in females in both the upper (-61.2%) and the lower body (-44.2%).”251- 252- in this study or generally? please provide reference

As suggested by reviewer 1, the period “Higher levels of relative…..compared to women.” Has been removed.

264- 267- provide more specific information to section participants

Amended

You have to provide all limitations, idea for future research and practical applications at the end of the study.

The following limitations have been included in the manuscript: “A limitation of the present investigation is that lifting technique and muscle activation were not assessed and sex differences cannot be accounted for lifting strategies. Another limitation may consist in the use of a single muscle thickness to adjust the 1RM of multi-joint exercises. In addition, strength and conditioning programs followed by the participants in the years preceding this study, likely differ in several variables (e.g., number of upper and lower body exercises, training frequency, periodization model etc.), and may have deeply influenced both performance and muscle morphology.”

Reviewer 3 Report

Overall

  • Please reduce the number of abbreviations. The manuscript and especially the abstract just require an enormous memory effort to remind each of them.
  • Please consider using “women” and “men” as nouns, and “female” and “male” as adjectives.

Introduction

  • Line 46: “gender differences between sexes”…what did the Authors mean?
  • The 2nd paragraph is unclear: too many new topics are introduced, without a logical order and rationale.
  • 2nd and 3rd paragraph: muscle architecture encompasses different parameters. As such, the Authors should expand this topic to let me understand which kind of relationship occurs.
  • The aims of the study are not justified by the previous sections. Moreover, why athletes? And which kind of athletes?
  • Lines 58-60: please rephrase: “strength” cannot be “male or female”

Methods

  • Line 66: “the participants…”. Very same for many other sentences, in which “the participants” (of this study) are mentioned.
  • The CMJ procedures should be described, in a dedicated paragraph. The Authors may refer to Coratella et al., 2018, Hum Mov Sci.
  • Please add more details to describe the procedures of the muscle architecture assessment, and also put the description in a logical order. For example, please describe the subjects’ position for each muscle assessment, then where the scan was positioned.
  • Statistical analysis: please list the parameters that will be examined.

Results

  • Because of too many abbreviations, this section is very hard to follow. Additionally, please be sure that every dependent parameter has been analyzed, with a particular reference to table 3.

Discussion

  • The 1st paragraph is supposed to help the reader (and me!) to understand what actually happened. In this case, this seems to be incomplete.
  • Given that the results section and the first paragraph do not provide a clear explanation of the results, I did not proceed with the rest of the discussion. I’ll do it gladly, once these concerns have been solved.

Author Response

Rew 3.

 The authors would like to thanks the Reviewer 3 for his suggestions and comments to improve the overall quality of the manuscript.

  • Please reduce the number of abbreviations. The manuscript and especially the abstract just require an enormous memory effort to remind each of them.

Amended. Abbreviations and acronyms have been reduced.

  • Please consider using “women” and “men” as nouns, and “female” and “male” as adjectives.

Amended.

Introduction

  • Line 46: “gender differences between sexes”…what did the Authors mean?

Amended. “Between sexes” has been removed.

  • The 2nd paragraph is unclear: too many new topics are introduced, without a logical order and rationale.

The second paragraph has been changed as follows: “Despite no significant differences between genders in muscle fibers number were reported [6], a qualitative difference in muscle tissue, such as a higher concentration of glycolytic enzymes and greater proportion of fast type muscle fibers [12], may explain the disparity in strength. Glycolytic capacity [12,13], as well as the muscle area occupied by fast type fibers, indeed, have been reported to be greater in male than in female individuals [14]. The gender difference in power may be also influenced by anthropometric and task-specific factors as well as morphological characteristics of muscles. More recently however, Perez-Gomez et al. [15], found similar power output normalised to muscle mass of the lower limbs in males and females. Maximal strength and power are influenced by many neuromuscular factors including muscle morphological characteristics such as muscle thickness, pennation angle and fascicle length [16,17]. Recently, moderate correlations have been reported between maximal isometric force expressed at the mid shin pull and muscle architecture of the vastus lateralis in resistance-trained individuals [16]. These parameters of muscle architecture, may be influenced by resistance training in both male and female athletes [18,19].”

  • 2nd and 3rd paragraph: muscle architecture encompasses different parameters. As such, the Authors should expand this topic to let me understand which kind of relationship occurs.

The following periods have been included in the section: “Recently, moderate correlations have been reported between maximal isometric force expressed at the mid shin pull and muscle architecture of the vastus lateralis in resistance-trained individuals [16]. These parameters of muscle architecture, may be influenced by resistance training in both male and female athletes [18,19].

  • The aims of the study are not justified by the previous sections. Moreover, why athletes? And which kind of athletes?

The aims have been changed as follows: “The aim of this study was to compare male vs. female athletes in strength and power performance relative to body mass, lean body mass and muscle architecture. A second aim of the present study was to assess the relationships between muscle architecture and maximal strength in resistance trained athletes competing in sports with elevated strength components.”

  • Lines 58-60: please rephrase: “strength” cannot be “male or female”

The perido has been changed as follows: “The aim of this study was to compare male vs. female athletes in strength and power performance relative to body mass, lean body mass and muscle architecture.”

Methods

  • Line 66: “the participants…”. Very same for many other sentences, in which “the participants” (of this study) are mentioned.

Amended

  • The CMJ procedures should be described, in a dedicated paragraph. The Authors may refer to Coratella et al., 2018, Hum Mov Sci.

Amended. The description and the suggested reference have been included in the manuscript: “They were required to keep their hands on their hips and to maximize their jump height, as previously described by Coratella et al. (..). They were also asked to perform 3 jumps with a recovery time of 3 minutes between the attempts. Peak power (CMJP) was calculated by the jump height and the participant’s body mass using the following equation: Peak Power = 60.7 x jump height + 45.3 x body mass – 22.055”

  • Please add more details to describe the procedures of the muscle architecture assessment, and also put the description in a logical order. For example, please describe the subjects’ position for each muscle assessment, then where the scan was positioned.

Amended. Further details about the participants body position during the assessments of muscle morphology were included.

  • Statistical analysis: please list the parameters that will be examined.

Statistical Analysis section has been changed as follows: “A Shapiro-Wilk test was used to test the normal distribution of the data. A relative reliability index (intra-class correlation coefficient, ICC) was used to examine the level of agreement between the attempts performed in each assessment. Absolute reliability (standard error of measurement, SEM) was used to define the extent to which a score varies on test-retest measurements. Independent sample t tests were used to compare the mean values of body composition and muscle architecture variables between groups. In addition, Edge’s g effect size (ES), and 95% confidence intervals (CI) were reported. Analysis of covariance (ANCOVA) was used to compare results of examined parameters between sexes while holding BM, LBM, or MT, constant [12]. For effect size, the partial eta squared 2) was reported, and according to Stevens [30], 0.01, 0.06, and 0.14 represent small, medium, and large effect sizes, respectively. Pearson’s product moment correlations were used to examine selected bivariate relationships. According to Mukkaka et al. [31], correlation coefficients (r) of 0.3, 0.5, 0.7, and 0.9 were interpreted as low, moderate, high, and very high, respectively. Significance was accepted at an alpha level of p ≤ 0.05, and all data are reported as mean ± SD. All analyses were performed using IBM SPSS, version 25.”

Results

  • Because of too many abbreviations, this section is very hard to follow. Additionally, please be sure that every dependent parameter has been analyzed, with a particular reference to table 3.

As suggested by the reviewer, acronyms have been reduced and re-defined. In addition, every dependent parameter have been checked.

Discussion

  • The 1st paragraph is supposed to help the reader (and me!) to understand what actually happened. In this case, this seems to be incomplete.

The first paragraph has been changed as follows: “The aim of the present investigation was to compare male and female resistance trained athletes in absolute and relative strength and power performances. Firstly, women were weaker than men in maximal strength expressed at bench press (-59.2%), squat (--57.2%), deadlift (-56.3%) and mid-shin pull (MSP, -53.2%). In addition, lower levels of power were detected in females in both the upper (-61.2%) and the lower body (-44.2%). This is consistent with previous studies [5,6] that reported similar differences between men and women in the upper body. The same authors however, reported that women were only 27% weaker than men in lower body strength.”

  • Given that the results section and the first paragraph do not provide a clear explanation of the results, I did not proceed with the rest of the discussion. I’ll do it gladly, once these concerns have been solved.

Thank you

Round 2

Reviewer 1 Report

PLEASE SEE ATTACHED DOCUMENT. 

Author Response

REVIEW REPORT MDPI

GENERAL COMMENTS:

The authors would like to say thanks again to the reviewer.

Specific Comments

Abstract Section

Please clarify the study aims. Particularly because you report two different ones (please see introduction and abstract).

Amended. The aim has been changed as follows: “The aim of this study was to compare male vs. female athletes in strength and power performance relative to body mass (BM) and lean body mass (LBM) and to investigate the relationships between muscle architecture and strength in both genders”

Introduction

Please revise the study aims, authors must be consistent throughout.

Amended

Lines 26-28. Both sentences are somewhat confusing and repetitive. Please merge or clarify.

The periods have been merged as follows: “Women athletes are known to be less strong and powerful than equally trained men [1], muscle strength of women indeed, is typically reported in the range of 40 to 75% of that of men [2]; women are also known to be less powerful than equally trained men. [3]. “

Lines 34-39. This sentence is hard to follow, maybe too long. Please improve wording.

Amended

Line 59. Therefore, the aim of the present study…

Amended

Secondary aim, still not clear to me what authors mean with ‘elevated strength components’. Clarify.

“Elevated strength components” has been replaced with: “in resistance trained athletes of both genders competing in sports with high expressions of strength and power. “

Methods

If authors consider that mean power can be easily calculated knowing the load lifted, please provide how was this performed. This must clear for the general reader, and to anyone wishing to replicate your study.

The following equation has been included in the section: “Mean Power (W) = repetition mean speed (m·sec−1) x force (N).”

Statistical Analysis

Results Section

In this section, authors don’t really need to replicate what is already in the tables (especially regarding correlations). Please be brief as this section is very hard to follow as it stands.

Amended. The section relative to correlations has been reduced.

Discussion Section

Lines 292-293. ‘women were weaker than men in maximal strength expressed’. Please reword to something like: ‘women had lower maximal strength values when compared to men…’ I suggest to avoid the word ‘weaker’, please revise throughout. Same with ‘sex’ and/or ‘sexes’ replace with ‘men and women’ or ‘both genders’…

Amended

Lines 343-344. VLFL abrev. has already been clarified in the methods section, also MSP and CMJP… Please revise throughout…

The authors understand the reviewer’s query. However, Rev 3 requested to define again the acronyms and to reduce abbreviations in the discussion.

Thank you

Reviewer 3 Report

The Authors have notably improved the manuscript. 

However, the reference list used here appears incomplete, and in some cases, very old articles were cited, ignoring very recent publications. For example, both within the introduction and discussion, muscle architecture was shown to correlate with high-velocity isokinetic contractions (Coratella et al., 2018 Hum Mov Sci), peak-power and time to peak power in cycling maximal test (Coratella et al., 2020 Res Quarterly Exerc Sport), and the rate of force development (Coratella et al., 2020 J Sci Med Sport). These are clearly in the spirit of the manuscript, and I believe such aspects should be mentioned.

Author Response

Rew 3.

Thank you again for your suggestions.

The following period has been included in the introduction: “ In addition, significant correlations were detected between muscle architecture of the vastus intermedius and the late phase of the rate of force development at isometric leg extension [18]. Further studies also reported significant correlations between muscle architecture of the vastus lateralis and peak power and time to peak power in an all-out Wingate test [19].” The following references have been included:

Coratella, G.; Longo, S.; Borrelli, M.; Doria, C.; Cè, E.; Esposito, F. Vastus intermedius muscle architecture predicts the late phase of the knee extension rate of force development in recreationally resistance-trained men. J Sci Med Sport 2020, 23: 1100-1104.

Coratella, G.; Longo, S.; Rampichini, S.; Limonta, E.; Shokohyar, S.; Valentina Bisconti, A.; Cè E.; Esposito, F. Quadriceps and gastrocnemii anatomical cross-sectional area and vastus lateralis fascicle length predict peak-power and time-to-peak-power. Res Q Exerc Sport 2020, 91: 158-165.

Thank you